# The Alteration of Circulating Invariant Natural Killer T, γδT, and Natural Killer Cells after Ischemic Stroke in Relation to Clinical Outcomes: A Prospective Case–Control Study

**DOI:** 10.3390/cells13161401

**Published:** 2024-08-22

**Authors:** Magdalena Frydrychowicz, Magdalena Telec, Jacek Anioła, Radosław Kazmierski, Hanna Chowaniec, Grzegorz Dworacki, Izabela Wojtasz, Wojciech Kozubski, Maria Łukasik

**Affiliations:** 1Department of Immunology, Poznan University of Medical Sciences, 60-806 Poznan, Poland; hchowaniec@ump.edu.pl (H.C.); gdwrck@ump.edu.pl (G.D.); 2Department of Neurology, Poznan University of Medical Sciences, 60-355 Poznan, Poland; mtelec@ump.edu.pl (M.T.); rkazmierski@ump.edu.pl (R.K.); wkozubski@ump.edu.pl (W.K.); mlukasik@ump.edu.pl (M.Ł.); 3L. Bierkowski Hospital, 60-631 Poznan, Poland; jacek.aniola@szpitalmswia.poznan.pl; 4Department of Neurology, Collegium Medicum, University of Zielona Gora, 65-417 Zielona Gora, Poland; 5Medicover, 61-894 Poznan, Poland; izabela.wojtasz@medicover.pl

**Keywords:** ischemic stroke, invariant NKT cells, γδT cells, natural killer cells

## Abstract

The adaptive response occurs only after 7–10 days of antigen presentation. Nevertheless, the autoreactive T cells infiltrate the stroke lesion within the first 48 h. Thus, we hypothesized that the unconventional lymphocytes as invariant natural killer T cells (iNKT) and γδT cells that share immediate innate and delayed adaptive response features are involved in acute stroke pathophysiology. We assessed prospectively the quantity of circulating iNKT cells, γδT cells, and NK cells with flow cytometry in 52 subjects within three months after stroke, and we compared the results with those obtained in age-, sex-, and vascular risk factor-matched controls. We studied lymphocyte parameters regarding clinical outcomes, infarct volume, stroke-associated infection (SAI), and burden risk factors. The reduced number of circulating γδT cells and decreased percentage of the Vδ2 subset in the acute phase of stroke correlated with worse neurological status in the recovery phase. In subjects treated with thrombolysis and those who developed SAI, a lower percentage of γδT cells in the 90-day follow-up was observed. An increased percentage of iNKT cells in the acute and subacute phases of stroke was observed, and it was related to the worse clinical status. The circulating NK cells do not change temporarily or affect the outcomes after stroke. It seems that γδT cells play a long-lasting role in ischemic stroke, mainly related to the Vδ2 subset. The role of iNKT cells appears to be detrimental, especially in the acute and subacute phases of stroke. The effect of circulating NK cells on the outcome after stroke seems negligible.

## 1. Introduction

Ischemic stroke is one of the leading causes of disability and mortality worldwide, and the number of cases continues to rise [1,2].

Historically, the brain has been considered an immunologically privileged area due to the presence of the blood–brain barrier and lack of lymphatic drainage [3,4,5].

Recent studies indicate that ischemic stroke leads to the activation of a cascade of immune processes. Although the infiltration of inflammatory leukocytes is a well-described feature of human stroke, the perspective that the activation of the immune systems is a bystander phenomenon secondary to ischemic tissue damage has changed. Currently, the activation of the immune system is recognized as a major element in all stages of the pathophysiology of stroke, including long-lasting regenerative processes [6].

The innate response in the vascular compartment occurs within the first hours of an ischemic stroke and involves the activation of monocytes, neutrophils, and mast cells, as well as NK cells and the complement system [3,7,8,9]. Initial brain damage during ischemia involves the disruption of the blood–brain barrier (BBB) and signaling to increase the expression of adhesion molecules on endothelial cells, which increase diapedetic potential of neutrophils and monocytes [7,10].

In response to ischemia and reperfusion, NK cells enter the vascular compartment of the central nervous system indirectly and directly, interacting with monocytes and platelets, promoting prothrombotic mechanisms and activating the complement system [8].

The adaptive immune response is much more complex. According to the rules of the adaptive response, the activation of lymphocyte subpopulations caused by contact with an antigen-presenting cell usually occurs within 5–7 days from the trigger factor. However, in ischemic stroke, it is the case already within the first 24 h from the onset of clinical symptoms [11,12]. This is one of the reasons why the antigen-dependent activation of T cells after stroke is the subject of much controversy.

The explanation of the rapid adaptive immune response in a stroke might be the involvement of unconventional γδT lymphocytes and iNKTs, namely subpopulations that combine the features of both innate and adaptive response cells and are able to react without prior antigen presentation.

T cells are integral to the pathophysiology of stroke. The initial inflammatory cascade leads to T cell migration, which results in deleterious or protective effects mediated through CD4(+), CD8(+), γδT, and regulatory T cells. γδT cells constitute 1–10% of peripheral blood lymphocytes and are a phylogenetically old subpopulation, including the Vδ1 and Vδ2 subtypes. The majority of human γδT cells in peripheral blood are Vδ2 cells. γδT cells are activated directly by danger-associated molecular patterns (DAMPs) (recognition of molecules released suddenly in a stressful situation) and through natural killer receptors (NKRs) without the need for clonal expansion nor antigen presentation, which allows γδT cells to function in parallel with innate response cells. Vδ2 lymphocytes recognize phosphorylated metabolites coming from microbes or the host. On the other hand, Vδ1 lymphocytes localize mainly in epithelia and recognize lipids and glycolipids presented by the CD1d particle [1,11,13,14]. The proportions of Vδ1 and Vδ2 subtypes change with age. From about 10 years of age, Vδ2 dominates quantitatively, but as they get older, the predominance of Vδ2 over Vδ1 lymphocytes, especially in males, begins to decrease. Within a few hours of activation and without prior systemic expansion, γδT cells can produce high concentrations of interferon gamma (IFN-γ), interleukin-17 (IL-17), interleukin-13 (IL-13), tumor necrosis factor-alfa (TNF-α), and granzymes [15,16]. Participation in the above processes makes the γδT cell subpopulation a bridge between the innate and adaptive immune response. Experimental works suggest a negative impact of γδT cells on the course and prognosis of ischemic stroke, which could be caused by the synthesis of pro-inflammatory IL-17A promoted by γδT cells, which stimulates the recruitment of neutrophils and their migration to the ischemic zone [6,17,18,19,20]. In patients with ischemic stroke, the presence of γδT cells was confirmed in the meninges, while in animal models, these cells were also visible in the penumbra, and both their absolute number and percentage increased significantly on the third day of acute ischemia. Moreover, reducing the number of γδT cells led to a reduction in the ischemic area and had a beneficial effect on the final clinical condition [17,19,20].

NKT cells constitute approximately 0.2% (0.01–1%) of the T cell population. In addition, it has been observed that unconventional invariant NKT lymphocytes (iNKT) are found among the various T cell populations recruited to the brain after stroke. The iNKT is a subpopulation that combines features of innate and acquired responses and does not require prior antigen presentation to function [5]. After ischemic stroke, iNKT lymphocytes are involved in stroke-induced immunosuppression. Studies show that they have the ability to respond rapidly to acute cerebral ischemia and move to the stroke zone from a distant location, such as the liver. Furthermore, it has been shown that the larger the stroke focus, the greater the activation of peripheral iNKTs [2,21]. Animal studies have shown that activated iNKT cells play a role in brain damage and accumulate in the ischemic area, increasing brain swelling and exacerbating neurological deficits. Stroke has also been shown to alter the function of iNKT cells, which are critical in regulating antimicrobial defense. In addition, it has been discovered that reversing stroke-induced impairment of iNKT cells restores normal immune function and reduces post-stroke infection, thus providing a pathway to reduce the leading cause of death in stroke patients [12,22,23].

NK cells are a specific population of immune cells that cross the blood–brain barrier during stroke and bridge the gap between the nervous and immune systems. In addition, NK cells are associated with post-stroke inflammation, immunosuppression, and infections [8]. In the early stages of the response to stroke, NK cells are redistributed and their number in the peripheral blood is reduced. However, their number within the parenchyma increases [24,25]. Recent studies indicate the involvement of NK cells in post-stroke immunosuppression and stroke-related infection. It has been shown that the number of circulating NK cells in the first hours of stroke is higher in those developing infection than in those without infection. This suggests that NK cells may be used as a harbinger of post-stroke infection. However, the results of quantitative assessment of NK cells in the peripheral blood of stroke patients are inconclusive, which may be related to individual variability, clinical status, and the presence of complications [26].

Based on current knowledge from experimental and clinical studies and considering the unexpectedly rapid adaptive immune response in a stroke, we hypothesized that the cells sharing both innate and adaptive response features are involved in acute stroke immunology. Since available data on the role of iNKT and γδT cell subpopulations and NK cells in ischemic stroke and recovery are sparse and often contradictory, many authors emphasize the need to expand existing knowledge, especially in clinical settings.

Thus, we aimed to prospectively assess the absolute number and the percentage of circulating non-conventional lymphocytes and NK cells, and the results were related to clinical data such as neurological status, stroke lesion volume, and treatment.

## 2. Materials and Methods

### 2.1. Study Participants

#### 2.1.1. Inclusion/Exclusion Criteria 

Inclusion criteria were as follows: (1) ischemic stroke, (2) age over 40, and (3) symptom onset no more than 24 h prior to clinical evaluation and blood sampling. The exclusion criteria were (1) subarachnoid and intracerebral hemorrhage, (2) cancer, (3) autoimmune diseases, (4) liver failure and kidney failure, (5) hematological disorders, (6) peptic ulcer disease, (7) alcohol and/or drug abuse, (8) history of infection and/or the use of antibiotics and/or immunosuppressants and/or steroids within the preceding 3 months. 

#### 2.1.2. Stroke Patients

Study participants were consecutively recruited among patients of the Department of Neurology and Cerebrovascular Disorders of Poznan University of Medical Sciences. The baseline group included 161 patients who presented with suspected stroke to the Stroke Unit between November 2018 and May 2019. The subjects were prospectively screened for inclusion in this study. A detailed flowchart has been published previously [27] (Appendix A). 

As a result of the inclusion criteria, 70 patients were pre-qualified for this study. Also, patients initially included in this study with a transient ischemic attack (TIA) were excluded if the symptoms resolved within 24 h and no lesions were visualized on magnetic resonance imaging (MRI) with diffusion-weighted imaging (DWI). The final study cohort consisted of 52 (25 women and 27 men) patients diagnosed with ischemic stroke, confirmed based on radiological evidence in cranial computed tomography (CT) scans and/or cranial MRI at admission. The ischemic stroke was diagnosed according to World Health Organization (WHO) stroke criteria. 

#### 2.1.3. The Clinical Assessment and Laboratory Investigations

Clinical assessment including physical and neurological examination was performed in the acute phase of stroke, on days 1 (D1) and 3 (D3), in the subacute phase on day 10 (D10), and 90 ± 3 days (D90) after stroke (convalescent phase). The subjects were assessed on the National Institutes of Health Stroke Scale (NIHSS) (Appendix A). The etiology of the stroke was classified according to the TOAST classification [28]. The standard diagnostic measures included blood pressure and height and weight assessments to calculate the Body Mass index (BMI, person’s weight in kilograms divided by the square of height in meters). The clinical evaluation was supplemented by laboratory investigations including blood count with the automatic smear test, coagulation, biochemical and urine tests, electrocardiography (ECG), chest radiograph, color Doppler duplex ultrasonography, transcranial Doppler ultrasonography (TCD), and transthoracic +/− transoesophageal echocardiography. Magnetic resonance imaging (MRI) was performed twice: within 24 h from the symptom onset and 90 ± 3 days after stroke. The ischemic lesion volume was measured according to the previously described protocols [27]. All stroke subjects were assessed for SAI on day 10 after stroke. SAI was defined as (1) body temperature > 37.8 °C in a patient with symptoms suggestive of infection and/or (2) white blood cell (WBC) count > 11,000/mL or <4000/mL and/or (3) inflammatory lesions in the chest X-ray and/or (4) blood or urine culture positive for a pathogen and/or (5) antibiotic or chemotherapeutic therapy, all within 7 days of the onset of stroke symptoms.

#### 2.1.4. The Disease Controls

The disease controls (DCs) included 34 age-, sex-, and vascular disease risk factor-matched patients. The controls underwent the same laboratory tests and procedures as stroke patients, except for chest radiography, neuroimaging, TCD, and echocardiography, and blood samples were taken only once. Control subjects with symptoms, signs, and abnormalities in laboratory tests suggesting acute/chronic inflammation were not included in this study.

### 2.2. Flow Cytometry

To detect the absolute number and percentage of circulating iNKT, γδT, and NK cells, flow cytometry combined with counting was used. In the stroke group, blood samples were obtained at 1, 3, 10, and 90 ± 3 days after stroke onset. In the control group, blood was drawn once. Cell immunophenotyping was performed by using the direct fluorescence method. The appropriate fluorochrome-labeled monoclonal antibodies (Appendix A) were added to the cytometric tubes according to the manufacturer’s instructions. In total, 100 µL of peripheral blood previously mixed with EDTA in vacutainer tubes was added to the test tubes. Samples were mixed gently by a vortex and incubated for 15 min at room temperature and protected from light. After this time, 500 µL lysis buffer (Becton Dickinson, Franklin Lakes, NJ, USA) was added to each test tube and the tubes were incubated for 10 min. Lysis was stopped by adding a phosphate-buffered saline (PBS, Roche, Germany) solution. Samples were centrifuged at 1500 rpm for 4 min at room temperature to separate residual lysed erythrocytes, plasma protein, and other elements of blood form leukocytes. The supernatants were discarded. This step was repeated once again. The cell pellets were resuspended in 200 µL of PBS. The samples were then acquired using a FACS CANTO II flow cytometer (Becton Dickinson). The obtained results were analyzed with the FACS Diva software version 6.2 (Becton Dickinson). For each examined antibody, the percentages of positive cells were determined. 

Lymphocytes were identified based on cell characteristic properties in the forward (FSC) and side (SSC) scatter. For additional analyses, for NK cells, gates were restricted to CD3−, CD3− CD56+, and iNKT subpopulations that were identified in the system FSC/SSC and based on the expression of the CD3+, CD3+ CD56+, and CD3+ CD56+ Vα24-Jα18+. γδT cells were defined based on the expression of CD3/γδTCR, and Vδ1 or Vδ2 subtypes were defined taking into account the coexpression of the corresponding δ chain fragments. The percentage of γδT cells was determined relative to the CD3+ lymphocyte population, and the percentage of Vδ1 and Vδ2 subpopulations was determined relative to γδT cells.

### 2.3. Statistical Analysis

Statistical analyses were conducted using the STATISTICA software version 13.0 (TIBCO Software Inc. (2017)), and GraphPad Prism version 8.0.1 (GraphPad Software, San Diego, CA, USA). The sample size was evaluated a priori using standard statistical criteria for the estimation of sample size and statistical power. In order to reveal at least 30% difference between studied groups, with statistical power of at least 90%, and a significance of at least 1% taking into account that the ratio of case-to-control sample size equals 1, the estimated minimum sample size is 34 stroke subjects and 34 matched controls. The normal data distribution was tested with Shapiro–Wilk and Kolmogorov–Smirnov tests. Data showing a non-normal distribution are presented as median values and interquartile range values and were analyzed with non-parametric methods. Data with normal distribution were presented as means ± SD. The parametric Student’s *t*-test, ANOVA, and post hoc Tukey test were used for comparisons of normally distributed data. Multiple comparisons within the stroke group were performed using the Wilcoxon signed-rank test with Bonferroni correction; thus, differences were assumed to be significant at *p* < 0.01. The Mann–Whitney U test was used to compare data between patients and the control group. Categorical data were compared by the chi-squared test or Fisher’s exact test where appropriate. The Spearman rank correlation test was used to test for possible relationships between the studied parameters. A multiple linear regression analysis was implemented to assess an independent association between iNKT cells and clinical outcome. Where not indicated otherwise, a *p* value < 0.05 was assumed to be significant. 

## 3. Results

### 3.1. Characteristics of Study Participants 

A total of 52 patients with ischemic stroke as well as 34 vascular disease controls were recruited in this study. The patients were evaluated on days 1, 3, 10, and 90. One patient died on day 7. On day 90, thirty-three patients were evaluated—eight patients refused to attend the last follow-up visit, four were unable to attend due to disability and dependency, three were undergoing inpatient rehabilitation at the time, and three patients had moved to another province. 

The mean age of patients and the control group was 69 ± 12 and 68 ± 12 years, respectively: 48.1% of patients were female, while the percentage of female participants in the control group was 38.2%. There were no significant differences in age and gender between patients and the control group (*p* > 0.05). There were no significant differences between studied groups in the proportion of smoking, hypertension, hyperlipidemia, ischemic heart disease, and diabetes. In the stroke group, more patients suffered from atrial fibrillation, and they were treated with oral anticoagulants (*p* < 0.05). 

The demographic and clinical characteristics of all study participants are summarized in Table 1.

The median time from stroke onset to blood sampling on D1 was 17 h (12–22). The peripheral blood leukocyte count (WBC), monocytes, erythrocytes, and platelets did not change significantly over time after the stroke. The highest WBC values were recorded at day 3 and the lowest at day 90 after the stroke; however, they were not statistically significant. The absolute number and the percentage of lymphocytes on post-stroke days 1, 3, and 10 were lower than those observed in controls, and on day 1, the percentage of lymphocytes was significantly lower than that measured on day 90 (Table 2).

### 3.2. Quantitative Assessment of iNKT, γδT, and NK Cells in Various Phases of Stroke

#### 3.2.1. iNKT Cells

As shown in Table 2, the absolute number of iNKT lymphocytes on D3 was significantly higher than in the control group. The percentage of iNKT cells in relation to the CD3^+^ lymphocyte population in patients on D1, D3, and D10 was significantly higher than in controls [D1: 0.9% (0.6–1.8), *p* = 0.03; D3: 0.9% (0.7–1.8), *p* = 0.04; D10: 0.9% (0.6–1.8) vs. CG: 0.5% (0.5–1.1), *p* = 0.02] (Figure 1). The values of the evaluated iNKT cell parameters did not change over the time elapsed since the stroke.

There were no differences between the absolute number of iNKT lymphocytes on D1, D10, and D90 and the value observed in controls, nor between the percentage of iNKT cells on D90 and the value in the control group.

#### 3.2.2. γδT Cells

The absolute numbers of γδT cells on D1 and D10 were significantly lower than in the control group (Table 2, Figure 2). 

The percentage of γδT cells in relation to the CD3^+^ lymphocyte population on D1, D3, D10, and D90 did not differ from the values observed in the control group and did not change significantly in the studied time points (Figure 3).

The absolute values and the percentage of Vδ1 lymphocytes in the γδT cell population in patients on D1, D3, D10, and D90 did not differ from the values obtained in the control group and did not change significantly in the time that had passed since the stroke. The percentage of Vδ2 lymphocytes in the γδT lymphocyte population in patients on D1, D3, and D10 was significantly lower than that observed in the control group [D1: 8.0% (5.3–15.6), *p*= 0.037; D3: 8.1% (3.6–16.5), *p* = 0.016; D10: 7.5% (5.3–17.7) vs. CG: 13.7% (6.8–20.1), *p* = 0.046], but did not change significantly in the time since the stroke. No significant differences in the absolute number of Vδ2 lymphocytes between the stroke group and controls at any studied time points were noted (Table 2, Figure 4a,b).

#### 3.2.3. NK Cells

The absolute numbers and percentage of NK cells in relation to the lymphocyte population in patients on D1, D3, D10, and D90 did not differ from the numbers obtained in the control group, and also did not statistically significantly change in the time elapsed since the stroke (Table 2, Figure 5).

### 3.3. Association of iNKT, γδT, and NK Cells with Clinical Status and Infarct Volume

There were significant differences between the clinical status of the patients on D1 and D3 and subsequent time points (D10 and D90). The median score on NIHSS was the highest on D1 [7 (4–11)] (the worst neurological status) and the median score on NIHSS was significantly lower on D10 [2 (0–5); *p* < 0.0001] and D90 [1 (0–4); *p* = 0.001], which indicates an improvement in the clinical condition. We found the percentage of the iNKT subpopulation on D1 and D10 to be positively associated with the NIHSS score on D1 (rS = 0.29, *p* < 0.05; and rS = 0.41, *p* < 0.01, respectively). Thus, in the acute phase of the stroke, a higher percentage of circulating iNKT cells correlates with worse clinical status. However, in the multiple regression analysis including other confounding variables such as age, infarct volume, and systolic blood pressure at admission, the percentages of iNKT cells on D1 were no longer independently correlated with neurological status. Moreover, there was no significant correlation between the iNKT subset and the clinical severity on D3 and D90.

There was a negative correlation between the percentage of γδT cells on D10 and the clinical outcome on D90 assessed in the NIHSS (rS = −0.61; *p* < 0.001). The clinical condition on D1 correlates negatively with the percentage of Vδ1 on D90 (rS = −0.4; *p* < 0.05). This means that a higher percentage of circulating γδT cells in the subacute phase is associated with better clinical status in the convalescence phase. However, a better clinical condition in the acute phase of stroke is associated with a higher percentage of circulating Vδ1 lymphocytes in the convalescence phase. The percentage of circulating Vδ2 did not correlate with neurological status at any studied time points. 

There were not any associations between the infarct volume on D1 or D90 and the percentage of circulating γδT, iNKT, and NK cells at any studied time points after stroke. 

### 3.4. Stroke-Associated Infection 

SAI criteria were met by 25 patients (48%). Eleven patients were diagnosed with SAI on the first day after stroke. Twenty patients had SAI+ as early as day 3. On day 10 after stroke, criteria for the diagnosis of SAI were found in 25 patients. The most common infection was pneumonia (14 patients; 56%), followed by urinary tract infection (9 patients; 36%) and other infections (2 patients; 8%). The baseline characteristics of SAI+ and SAI− patients are shown in Table 3. 

On D90, the percentage of γδT cells was lower in the SAI+ group than in SAI− patients [1.5% (0.9–2.5) vs. 2.5% (2.0–5.4); *p* = 0.03] (Figure 6).

There were no significant quantitative differences in the percentage of NK cells and iNKT lymphocytes between SAI+ and SAI− patients.

### 3.5. Quantitative Assessment of iNKT, Tγδ, and NK Cells in Relation to the Risk Factors of Ischemic Stroke

In both stroke subjects and controls, we have found no differences in the absolute number and percentage of circulating iNKT, γδT, and NK cells depending on the burden of stroke risk factors such as hypertension, hyperlipidemia, atrial fibrillation, and smoking. There was no correlation between age and quantity of studied cells. 

On D10, we observed a lower percentage of γδT lymphocytes in diabetic patients than in non-diabetic ones [1.7% (0.9–2.2) vs. 3.0% (2.1–4.9), respectively; *p* = 0.01] (Figure 7).

### 3.6. Quantitative Assessment of iNKT, Tγδ, and NK Cells in Relation to the Thrombolytic Treatment

On D90, the percentage of γδT cells was significantly lower in stroke patients who received the thrombolytic treatment (rTPA+) as compared to those who were not treated with thrombolysis (rTPA−) [2.1 (0.9–2.3) vs. 2.8 (1.8–5.9); *p* = 0.025] (Figure 8). There were no quantitative differences between rTPA+ and rTPA− subjects in the percentage of circulating γδT in other studied time points as well as in the percentage of iNKT and NK cells on D1, D3, D10, and D90. 

## 4. Discussion

Ischemic stroke is one of the leading causes of disability and death among adults worldwide. The immune response triggered by ischemic stroke plays a crucial role in the pathogenesis of brain damage and contributes to tissue regeneration processes [29].

It has been shown that unconventional T lymphocytes, such as iNKT and γδT cells, play an important role in immunological regulation, acting mainly by bridging innate and acquired immune responses [29,30].

Unconventional T cells characterized by an invariant TCR have become increasingly recognized as a contributor to ischemic damage. These cells produce IL-17, which exacerbates neuroinflammation through the induction of Granulocyte-Colony Stimulating Factor (G-CSF) and other chemokines, which foster the recruitment of pro-inflammatory immune cells such as neutrophils into the brain [29,30].

Experimental studies confirm the presence of γδT lymphocytes in the penumbra area. Furthermore, in postmortem studies of stroke patients, the presence of γδT lymphocytes in the meninges was confirmed, and simultaneous reduction in the number of these cells in peripheral blood was observed. This may indicate their involvement in central localization [6,18,31].

We demonstrated that patients with cerebral ischemia have significantly reduced numbers of γδT lymphocytes in the acute and subacute phases of stroke compared to the control group. Moreover, we also demonstrated reduction in the proportion of the Vδ2 subtype in the γδT lymphocyte population in the acute and subacute phases of stroke compared to the control group, while the Vδ2 subtype was quantitatively dominant in peripheral blood of adults. Our results correspond with the findings of Adamski et al., who demonstrated a reduction in circulating γδT lymphocytes in stroke patients compared to the control group while also showing increased activity of their product, IL-17A. γδT cells that secrete IL-17 have been characterized as ligand-naive and require activation by IL-23 [32]. 

Experimental studies, where a reduction in the number of γδT lymphocytes led to a reduction in the ischemic area and positively impacted the final clinical state of patients, confirmed the unfavorable, pro-inflammatory role of γδT lymphocytes in ischemic stroke, primarily due to the synthesis of pro-inflammatory IL-17A [22,31,33]. Furthermore, our study demonstrated that a lower percentage of γδT lymphocytes in peripheral blood in the subacute phase of stroke is associated with poorer clinical status in the recovery phase, which may confirm the hypothesis that a lower number of γδT lymphocytes in peripheral blood results from their migration to the stroke area and their unfavorable action in the central nervous system. It is known that dendritic cells, migrating very early to the ischemic area, are a source of IL-23, through which they stimulate the activation of γδT lymphocytes, encouraging them to synthesize pro-inflammatory IL-17 and intensive recruitment of neutrophils to the infarct area. This seems to be the basic sequence of the innate response, which tightly links γδT lymphocytes with neural tissue destruction.

Previous studies did not examine the impact of γδT lymphocytes on clinical status and neurological deficit. Our study results may indicate the importance of this subpopulation in the long-term immune response to ischemic stroke. In patients who have previously had an ischemic stroke, a statistically significant increase in the percentage of the Vδ2 in the recovery phase compared to patients without this risk factor in their history was demonstrated. In the presented study, a statistically significant reduction in their percentage in the subacute phase of stroke in patients with type 2 diabetes was shown compared to patients without diabetes. In one of the few existing clinical studies assessing the participation of γδT lymphocytes in diseases that are risk factors for stroke, a reduction in the number of γδT lymphocytes was demonstrated in people over 60 years old, with a history of ischemic heart disease, and especially in patients with hypertension [32]. It is known that γδT lymphocytes synthesize molecules with a confirmed role in vascular wall fibrosis and thus in the development of hypertension such as IL-17, IFNγ, TNFα, or CCL5. It is possible that the involvement of γδT lymphocytes in the pathology of these diseases, which are simultaneously risk factors for vascular diseases, explains the demonstrated correlation. Although the connection of the immune system with hypertension, vascular atherosclerosis, or diabetes is no longer surprising, the exact mechanisms and roles of specific cell populations are not well understood [34,35].

The conducted studies indicate that the role of γδT lymphocytes in stroke appears to be long-term, with adverse clinical effects visible both in the acute phase and several months after the stroke, and they may be related to the central activity of the Vδ2 subtype. Moreover, in the recovery phase, a significant decrease in the percentage of γδT cells was observed in patients who received thrombolytic treatment (rTPA+) compared to those who were not treated (rTPA−). Previous observations suggest a controversial but possible role of rTPa in damaging the blood–brain barrier, promoting neurotoxicity, and enhancing the inflammatory response, intensifying secondary brain damage [36]. The reduction in the percentage of Tγδ cells in the convalescent phase in both patient groups (SAI+ and rTPA+) indicates the significant role of γδ T cells in the long-term response to stroke.

As iNKT cells are known to directly regulate immune responses through their rapid and massive production of a wide range of cytokines, or indirectly through their regulation of other immune cell types, the precise role of iNKT cells in cerebral ischemia could be immensely complex and diverse [29,37].

In our study, a significant increase in the percentage of circulating iNKT lymphocytes in the acute and subacute phases of ischemic stroke compared to the control group was also observed. In the only clinical study, Wong et al. did not note a change in the number of iNKT in peripheral blood but confirmed a significant increase in iNKT cell activation during stroke, positively correlated with the severity of the neurological deficit and immunosuppression [2]. The role of iNKT lymphocytes in ischemic stroke was previously described in an experimental study on rodents by the same researchers [21]. Other experimental studies in this area only note the presence of NKT lymphocytes in the ischemic area, indicating the need to define their role in stroke [12,22,23].

The results of our study do not fully correspond with those obtained by Wong et al., but it should be emphasized that there are numerous methodological differences between the studies that do not allow for precise comparison. Wong et al. excluded patients with diabetes and a history of lymphopenia from the study. Such a criterion was not adopted in our study, and patients with diagnosed diabetes constituted nearly 20% of the study group. The control group also differed, which in our study consisted of individuals with recognized risk factors for vascular diseases, whereas Wong et al. created two control groups, one of healthy individuals and another of subjects hospitalized due to “stroke-like” symptoms, where neither stroke nor another specific nosological entity was ultimately diagnosed. The gating method for iNKT lymphocytes also differed. Nevertheless, Wong et al. suggest an adverse role of iNKT cells in stroke, similarly to our findings regarding correlation between a higher number of circulating iNKT lymphocytes and a greater neurological deficit and disability [23].

Although an increase in iNKT lymphocytes in peripheral blood in the early period of stroke seems to translate into a worsening of the clinical condition, the mechanism of this association remains unknown. Nevertheless, according to the results of the multiple regression analysis on the limited number of subjects, iNKT cells do not seem to be an independent factor for worse clinical outcome, and extensive cohort studies are needed to establish the actual effect of iNKT on neurological deficits.

Experimental studies also indicate the early presence of NK cells in the penumbra area and an increase in their number in the ischemic parenchyma, confirming their participation in enlarging this area by promoting neuronal necrosis [22,33,38]. NK cells infiltrate the brain as early as three hours after stroke, peaking three days after the ischemic event. Gan et al. demonstrated that NK cells were located in perivascular areas or ischemic penumbra in mice with transient middle cerebral artery occlusion (tMCAO), as well as in postmortem tissue of stroke patients. It is widely known that NK cells exert their pathogenic, inflammatory, and neurotoxic abilities through the secretion of IFNγ and perforin. A study conducted by Zhang et al. showed that NK cells participate in BBB disruption in an interferon gamma-induced protein 10 (IP-10)-dependent process. Increased amounts of interleukin-15 (IL-15) and chemokine CX3CL1 are secreted in the ischemic brain, with both being necessary for NK cell recruitment. Furthermore, IL-15 causes maturation and increases the cytotoxic function of NK cells [33,38].

In our study, no differences were found in the percentage of NK cells relative to the lymphocyte population in patients after stroke compared to the control group. No changes were found over time since the stroke. This result is consistent with those obtained by Vogelgesang et al., who did not find differences in the absolute number and percentage of NK cells in the lymphocyte population in patients after stroke compared to healthy individuals [39]. Similar results were obtained by Haeusler et al., Urra et al., Yan et al., and Xiao et al. [40,41,42,43]. 

De Raedt et al. demonstrated an increase in the number of NK cells in circulating blood, but only in the group of patients developing post-stroke infection symptoms, and the number of NK cells positively correlated with the ischemic area [26]. 

Additionally, they showed that better clinical status in the acute phase correlates with a higher percentage of NK cells in peripheral blood during the subacute phase of stroke. This suggests that in patients with better outcome, NK cell involvement at the ischemic area may be less, resulting in increased numbers in peripheral blood during the subacute phase. Previous studies indicated the harmful impact of NK cells in stroke progression, confirmed by functional studies showing reduced pro-inflammatory IFN-γ and perforin expression, potentially increasing susceptibility to post-stroke infections. Gan et al. and de Raedt et al. confirmed that higher circulating NK cells shortly after a stroke were associated with a higher incidence of SAI, suggesting a predictive value of NK cells for SAI [26,38]. Our study did not show such an association. The high inter-individual variability in circulating NK cell numbers, lacking pre-stroke baseline data, fundamentally excludes its use as an SAI predictor. Widespread NK cell infiltration into the ischemic area has also been described. Thus, despite many theoretical and experimental premises regarding the negative role of NK cells in stroke, clinical studies have not established whether this effect is due to local actions within the ischemic tissue or dysfunction leading to reduced infection defense.

Neither our study nor Yan et al.’s showed quantitative differences in NK cells between individuals with typical vascular disease risk factors and those without among both the stroke patients and controls [42]. This may indicate that the documented experimental influence of NK cell activation on developing conditions such as hypertension or diabetes, despite clinical study discrepancies, may not be as crucial in a sample of several dozen individuals [44].

Despite the growing number of immunological response observations during ischemic stroke in clinical settings, the exact role of γδT and iNKT lymphocytes and NK cells remains unclear. The precise role of these cell subpopulations in ischemic stroke is highly complex and varied. There are some limitations of our study. Undoubtedly, the major one is the limited sample size. It makes it impossible to assess whether the relationship between lymphocytic parameters and the clinical condition is independent or may result from the influence of other factors. Moreover, we evaluated the unconventional lymphocytes and NK cells only within one compartment, using cytometry as the only method, and our data are not supported indirectly by functional strategies.

## 5. Conclusions

Despite decades of promising experimental animal studies and clinical translations, the immune system’s involvement in stroke pathophysiology remains unclear. However, we are still at the beginning of this challenging but—as preliminary data show—significant and exciting journey to explain the sequence of events in the early stages of stroke that influence patients’ outcomes, including the recovery period. In summary, this is one of the few studies to characterize the circulating profiles of unconventional subpopulations of T lymphocytes and NK cells in patients with different stages of ischemic stroke. It seems that γδT cells play a long-lasting role in ischemic stroke that is particularly related to the Vδ2 subset. The reduced number of circulating γδT cells in the acute phase of stroke correlated with worse neurological status in the recovery phase and probably resulted from the cell shift to the brain compartment. The role of circulating iNKT cells appears to be detrimental especially in acute and subacute phases of stroke. The effect of circulating NK cells on outcome after stroke seems to be negligible.

## Figures and Tables

**Figure 1 cells-13-01401-f001:**
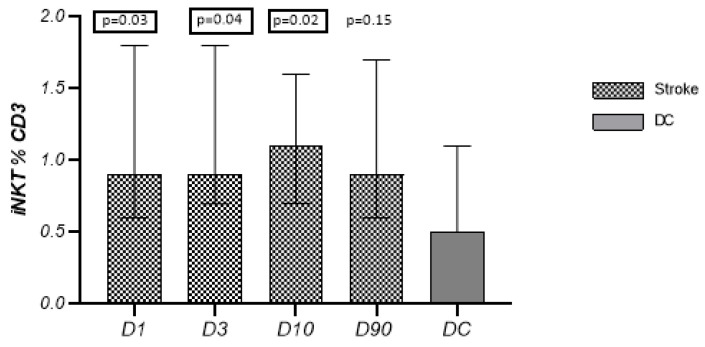
Temporal pattern of iNKT cells in subjects after stroke and disease control.

**Figure 2 cells-13-01401-f002:**
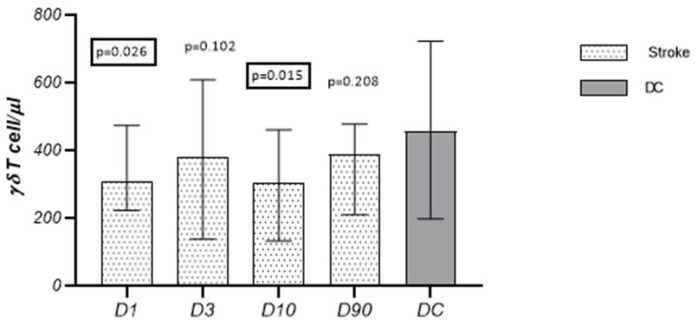
Temporal pattern of γδT cells in subjects after stroke and disease control.

**Figure 3 cells-13-01401-f003:**
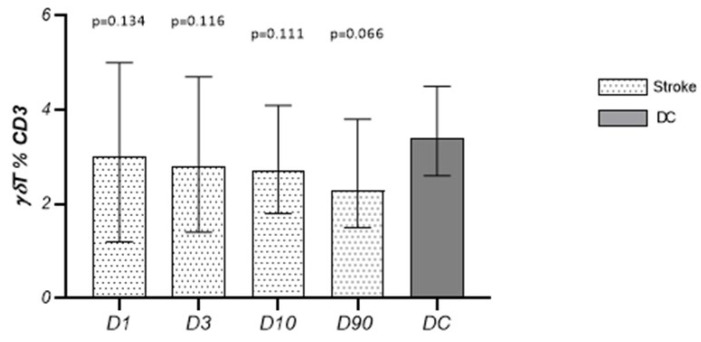
The percentage of γδT cells in relation to the CD3^+^ lymphocyte population on D1, D3, D10, and D90.

**Figure 4 cells-13-01401-f004:**
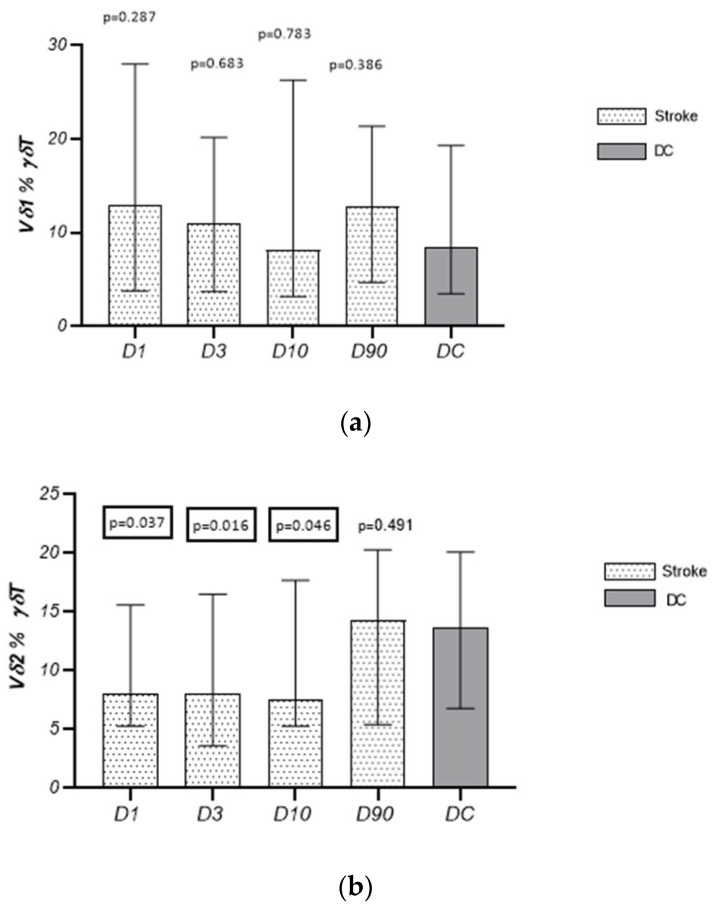
Temporal pattern of T Vδ1 (**a**) and Vδ2 cells (**b**) in subjects after stroke and disease control.

**Figure 5 cells-13-01401-f005:**
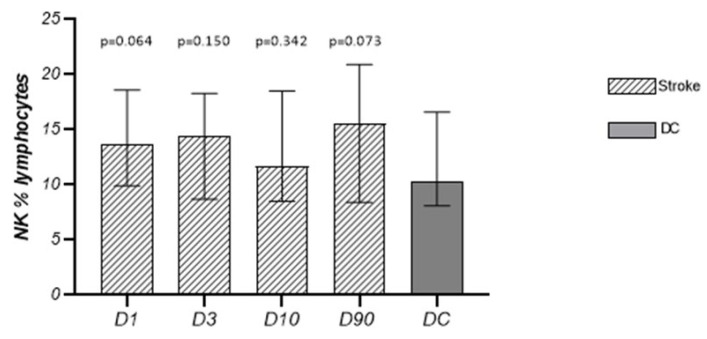
Temporal pattern of NK cells (%) in subjects after stroke and disease control.

**Figure 6 cells-13-01401-f006:**
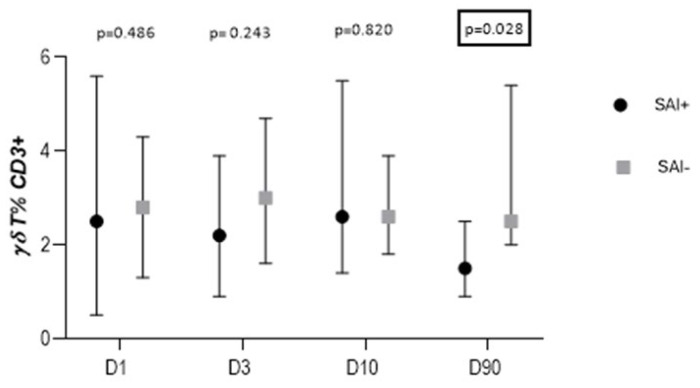
Percentage of γδT cells in patients with and without stroke-associated infection (SAI) on D90.

**Figure 7 cells-13-01401-f007:**
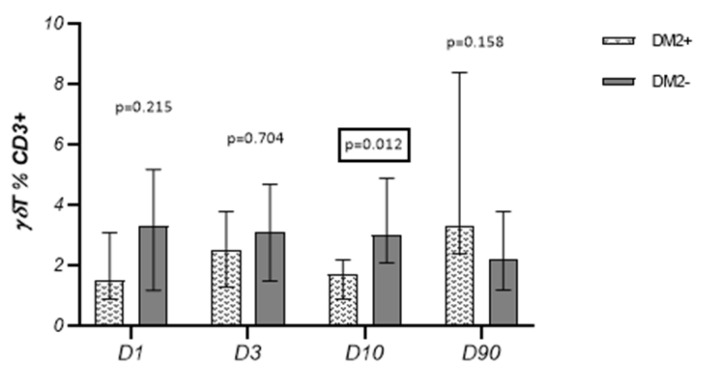
Percentage of γδT cells in patients with and without diabetes.

**Figure 8 cells-13-01401-f008:**
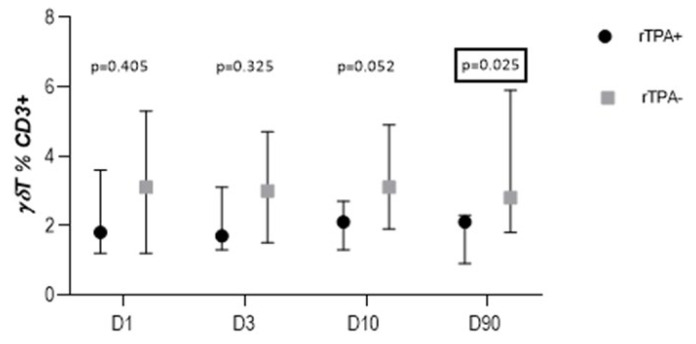
Percentage of γδT cells in patients after stroke treated with thrombolysis (rTPA+) and those without such a treatment (rTPA−).

**Table 1 cells-13-01401-t001:** Characteristics of the study cohort.

	Stroke Patients D1 N = 52	Disease Controls N = 34	*p*-Value
Age, years	69 (±12)	68 (±13)	0.66
BMI (kg/m^2^)	25.4 (23.6–28.4)	27.0 (25.1–31.1)	0.37
Females, n (%)	25 (48)	13 (38)	0.39
Hypertension, n (%)	42 (81)	30 (88)	0.56
Diabetes, n (%)	10 (19)	6 (18)	0.99
Ischemic heart disease, n (%)	19 (37)	14 (41)	0.82
Atrial fibrillation, n (%)	20 (39)	3 (9)	<0.01
Hyperlipidemia (%)	16 (31)	12 (35)	0.81
Smoking, n (%)	18 (35)	7 (21)	0.23
Treatment, n (%)			
Thrombolysis	16 (31)	-	
Mechanical thrombectomy	0	-	
Antiplatelet drugs	45 (87)	34 (100)	0.72
Anticoagulant	7 (14)	0	0.04
ACEI	20 (39)	17 (50)	0.37
Diuretics	23 (44)	12 (35)	0.51
CCB	22 (42)	17 (50)	0.51
β-blockers	9 (17)	8 (24)	0.58
ARB	7 (14)	1 (3)	0.14
Statins	14 (27)	15 (44)	0.11
Hypoglycemics	8 (15)	4 (13)	0.78
Insulin	3 (6)	3 (9)	0.68
Stroke etiology (TOAST classification), n (%)			
LVD	13 (25)	-	-
SVD	12 (23)	-	-
CE	22 (42)	-	-
OE	5 (10)	-	-
UE	0	-	-
Stroke location, n (%)			
TACI	6 (12)	-	-
PAC	22 (42)	-	-
POCI	13 (27)	-	-
LACI	10 (8)	-	-
Stroke lesion volume, mL			
Day 1	2.1 (0.3–11.2)	-	-
Day 90	1.0 (0.4–4.7)	-	-

ACEI, angiotensin convertase enzyme inhibitor; CCB, calcium channel blocker; ARB, angiotensin receptor blocker; LVD, large-vessel disease; SVD, small-vessel disease; CE, cardioembolic stroke; OE, stroke of other etiology; UE, stroke of unknown etiology; TACI, total anterior circulation infarct; PACI, partial anterior circulation infarct; POCI, posterior circulation infarct; LACI, lacunar infarct.

**Table 2 cells-13-01401-t002:** Prospective quantitative analysis of white blood cell subpopulations in stroke patients.

	Stroke D1 N = 52	Stroke D3 N = 52	Stroke D10 N = 51	Stroke D90 N = 33	Disease Control Group N = 34	pD1 vs. DC	pD3 vs. DC	pD10 vs. DC	pD90 vs. DC
**WBCs** × 10^3^/µl	7.4 (5.8–10.6)	7.6 (5.7–10.2)	7.2 (5.9–8.1)	6.8 (5.6–7.7)	6.8 (5.7–8.4)	0.12	0.28	0.82	0.45
**Lymphocytes** × 10^3^/µL	1.7 ± 0.9	2.0 ± 0.9	1.8 ± 0.7	1.9 ± 0.7	2.3 ± 1.2	0.008	0.014	0.024	0.07
**Lymphocytes** %	21.5 ± 11.6	25.3 ± 10.9	25.8 ± 10.2	28 ± 7.5	31.3 ± 10	<0.0001	0.018	0.029	0.16
**NK** (cells/μL)	183 (106–262)	188 (122–288)	182 (97.3–273)	181 (114–317)	171 (103–293)	0.884	0.682	0.923	0.504
**iNKT** (cells/μL)	7.7 (4.4–13.1)	10.1 (6.3–16.6)	8.0 (5.3–12.8)	7.3 (4.5–14.2)	7.0 (3.3–11.9)	0.534	0.045	0.380	0.769
**γδT** (cells/µL)	311 (224–476)	382 (140–611)	306 (135–462)	390 (211–480)	457 (200–724)	0.026	0.102	0.015	0.208
**γδT Vδ1** (cells/µL)	4.0 (1.6–10.3)	5.1 (1.3–10.7)	5.6 (1.8–9.1)	4.9 (2.9–12.1)	3.8 (2.2–9.1)	0.631	0.772	0.733	0.572
**γδT Vδ2** (cells/µL)	8.2 (3.9–16.8)	10.5 (4.1–26.0)	7.4 (2.9–16.9)	18.9 (3.7–32.1)	12.5 (4.9–22.5)	0.316	0.932	0.126	0.415

WBCs—white blood cells; lymphocytes—percentage of lymphocytes in leukocyte population (automatic blood smear); D1, D3, D10, D90—days after stroke.

**Table 3 cells-13-01401-t003:** Comparative characteristics of SAI+ and SAI− patients.

	SAI+ N = 25	SAI− N = 27	*p* Value
Age, years	73 ± 13	66 ± 12	0.24
Sex, M/F, n	12/13	14/12	0.76/0.83
Time from stroke onset to blood sampling on D1, hours	18.5 (12−22)	17 (12−21)	0.57
CRP D1, mg/L	7.5 (6.1−27.5)	3.6 (1.3−6.4)	0.004
CRP D3	17.3 (7.9−69.5)	3.5 (1.2−6.5)	<0.001
CRP D10	20.7 (2.8−45.1)	2.5 (1.3−5.1)	0.002
CRP D90	3.5 (1.4−5.3)	2.3 (1.1−3.8)	0.21
WBCs D1, ×10^3^/µL	10.6 ± 3.6	7.6 ± 2.5	<0.0001
WBCs D3	10.7 ± 5.3	6.8 ± 1.6	<0.00001
WBCs D10	9.1 ± 3.9	6.7 ± 1.5	0.002
WBCs D90	7.1 ± 2.0	7.0 ± 0.9	0.63
Lymphocytes D1, %	17.0 ± 10.2	25.4 ± 8.6	<0.001
Lymphocytes D3	16.8 ± 6.9	27.7 ± 8.7	<0.0001
Lymphocytes D10	19.1 ± 10.4	27.4 ± 8.7	0.006
Lymphocytes D90	27.1 ±7.3	28.9 ± 8.4	0.91
NK% CD3 D1, %	14.4 (10.2−20.2)	13.6 (9.9−17.5)	0.68
NK% CD3 D3	14.6 (11.6−20.1)	13.9 (8.1−17.0)	0.49
NK% CD3 D10	12.4 (9.4−17.3)	11.4 (7.8−18.6)	0.55
NK% CD3 D90	17.0 (13.5−21.4)	13.8 (8.1−20.9)	0.57
iNKT % D1, %	0.8 (0.5−1.9)	1.0 (0.6−1.8)	0.33
iNKT % D3	0.9 (0.7−1.8)	1.1 (0.7−1.7)	0.81
iNKT % D10	1.2 (1.0−1.7)	1.0 (0.5−1.6)	0.12
iNKT % D90	1.0 (0.7−1.4)	0.8 (0.6−1.8)	0.85
γδT % D1, %	2.5 (0.5−5.6)	2.8 (1.3−4.3)	0.48
γδT % D3	2.2 (0.9−3.9)	3.0 (1.6−4.7)	0.24
γδT % D10	2.6 (1.4−5.5)	2.6 (1.8−3.9)	0.82
γδT % D90	1.5 (0.9−2.5)	2.5 (2.0−5.4)	0.02
NIHSS D1, pts	9 (5−19)	3 (1−6)	0.001
NIHSS D3	13 (5−25)	2 (1−4)	<0.001
NIHSS D10	13 (2−20)	1 (0−2)	<0.001
NIHSS D90	4 (2−6)	1 (0−3)	0.15
Stroke volume D1, mL	5.6 (0.2−25)	2.8 (0.5−9.5)	0.45
Stroke volume D90	1 (1−25)	1 (0.1−2.7)	0.76

WBCs—white blood cells; lymphocytes—percentage of lymphocytes in leukocyte population (automatic blood smear); NK%CD3—percentage of NK in lymphocyte population (flow cytometry); iNKT%CD3—percentage of iNKT in lymphocyte population; γδT % CD3—percentage of γδT in lymphocyte population (flow cytometry); NIHSS—National Institutes of Health Stroke Scale; D1, D3, D10, D90—days after stroke.

## Data Availability

The data presented in this study are available in this article.

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
