# Peer review of "The Alteration of Circulating Invariant Natural Killer T, γδT, and Natural Killer Cells after Ischemic Stroke in Relation to Clinical Outcomes: A Prospective Case–Control Study"

_cells, 2024, doi:10.3390/cells13161401_

Round 1

Reviewer 1 Report

Comments and Suggestions for Authors

Alteration of circulating iNKT, γδT and NK cells after ischemic stroke in relation to clinical outcomes: a prospective case-control study

M&M
Inclusion criteria were as follows: ischemic stroke….

The authors didn’t mention what kind of treatment patients had received? Was it best medical treatment (BMT) or they had received rt-PA or mechanical thrombectomy was performed.
The type of stroke treatment has to be mentioned since it is known that rt-pa and MT also trigger certain inflammatory response. Moreover, the inflammatory status after MT appears to affect brain repair in the long run.

Study subjects
• line 125 and line126
The sentence should be corrected

RESULTS
• line 226 and 227
The sentence should be corrected

• Table 2.
The description of the Table 2. should be corrected: no NIHSS or mRS in the Table 2.
Furthermore, correct NIHSS - it is not National Institutes of Health Scale Score but the National Institutes of Health Stroke Scale

• lines 299-301
There were significant differences between the clinical status of the patients on D1 and at subsequent time points. The score was the highest on the NIHSS (the worst clinical outcome) on D3, D10, D90 the NIHSS score was lower, which indicates an improvement in the clinical condition.

Confusing - please clarify

• lines 303-305
The NIHSS score was significantly higher (worse clinical outcome) in SAI+ patients on D1, D3 and D10. On D90 the neurological status did not differ between SAI+ and SAI- groups.

The authors equalize HIHSS with clinical outcome on D1, D3 and D10. First of all, clinical outcome in clinical settings won’t be analysed of those days; furthermore, NIHSS score can be low but patient could suffer from serious posterior circulation stroke.
Stroke, 2021 - Previous studies have reported that >75% of patients with posterior circulation stroke present with a baseline NIHSS score of 0 to 5.

• lines 306-307
We found the percentage of iNKT subpopulation on D1 and D10 to be positively associated with the NIHSS score on D1 and the mRS score on D10.

Why to assess mRS on D10? There is no relevant clinical explanation for it, since patient will surely undergo rehabilitation.

Reviewer 2 Report

Comments and Suggestions for Authors

Thank you very much for conducting research on such an important and relevant topic today. I found your paper very interesting; however, I need to make a few comments:

**Abstract:** Please add the study's objective, as well as the study's conclusions.

**Introduction:** Detail the hypothesis and objective of the study at the end of the introduction.

**Materials and Methods:** Section 2.1 is very long; it should be rewritten in a more organized manner to improve its clarity. You can divide it into more subsections and explain the study procedure, the instruments used for measurement, and the intervention carried out. The sections should be logically and coherently organized to enhance visual order. The inclusion criteria need to be detailed.

Additionally, explain in more detail how the patient recruitment was done, where it took place, and how it was conducted. Was informed consent obtained from the participants? How was the group assignment carried out? You should indicate the approval of the ethics committee.

Including a flowchart explaining patient selection and the procedure conducted would provide greater clarity and understanding for the reader. 

Explain in more detail the scales used, such as the NIHSS, mRS, and BMI, including what they measure and their scoring systems.

**Results:** Use unified criteria to detail significance in Table 1. In the first lines, it is used in one way and then as p=... What does >0.99 mean for the diabetes variable? Is that sign > necessary? Review the value <0.0001 in Table 2. Is it correct? The decimals in Table 3 should be indicated with a point.

**Conclusions:** This section should be expanded, providing the main conclusions derived from this study.

Round 2

Reviewer 2 Report

Comments and Suggestions for Authors

The authors have responded to all suggestions made and the quality of the manuscript has been improved. Thank you